# Venom Proteomics of *Trimeresurus gracilis*, a Taiwan-Endemic Pitviper, and Comparison of Its Venom Proteome and VEGF and CRISP Sequences with Those of the Most Related Species

**DOI:** 10.3390/toxins15070408

**Published:** 2023-06-22

**Authors:** Tsz-Chun Tse, Inn-Ho Tsai, Yuen-Ying Chan, Tein-Shun Tsai

**Affiliations:** 1Institute of Wildlife Conservation, National Pingtung University of Science and Technology, Pingtung 912301, Taiwan; ivan44654a@gmail.com; 2Institute of Biological Chemistry, Academia Sinica, Taipei 11529, Taiwan; bc201@gate.sinica.edu.tw; 3Institute of Biochemical Sciences, National Taiwan University, Taipei 106319, Taiwan; 4Department of Biological Science and Technology, National Pingtung University of Science and Technology, Pingtung 912301, Taiwan; chanyuenying23@gmail.com

**Keywords:** snake venom proteomics, CRISP sequences, VEGF sequences, sequence alignments, *Trimeresurus gracilis*, *Ovophis okinavensis*, human health

## Abstract

*Trimeresurus gracilis* is an endemic alpine pitviper in Taiwan with controversial phylogeny, and its venom proteome remains unknown. In this study, we conducted a proteomic analysis of *T. gracilis* venom using high-performance liquid chromatography-tandem mass spectrometry and identified 155 toxin proteoforms that belong to 13 viperid venom toxin families. By searching the sequences of trypsin-digested peptides of the separated HPLC fractions against the NCBI database, *T. gracilis* venom was found to contain 40.3% metalloproteases (SVMPs), 15.3% serine proteases, 6.6% phospholipases A_2_, 5.0% L-amino acid oxidase, 4.6% Cys-rich secretory proteins (CRISPs), 3.2% disintegrins, 2.9% vascular endothelial growth factors (VEGFs), 1.9% C-type lectin-like proteins, and 20.2% of minor toxins, nontoxins, and unidentified peptides or compounds. Sixteen of these proteoforms matched the toxins whose full amino-acid sequences have been deduced from *T. gracilis* venom gland cDNA sequences. The hemorrhagic venom of *T. gracilis* appears to be especially rich in PI-class SVMPs and lacks basic phospholipase A_2_. We also cloned and sequenced the cDNAs encoding two CRISP and three VEGF variants from *T. gracilis* venom glands. Sequence alignments and comparison revealed that the PI-SVMP, kallikrein-like proteases, CRISPs, and VEGF-F of *T. gracilis* and *Ovophis okinavensis* are structurally most similar, consistent with their close phylogenetic relationship. However, the expression levels of some of their toxins were rather different, possibly due to their distinct ecological and prey conditions.

## 1. Introduction

Taiwan is a mountainous island, with two thirds of its territory covered by mountain forests. Today, alpine organisms worldwide face various threats such as climate change, air pollution, and human development; thus, research on alpine organisms is urgently needed. Among the more than 200 mountains in Taiwan higher than 3000 m, some glacial refuges and relic species have considerably high conservation value [1,2,3]. *Trimeresurus gracilis* Oshima, 1920 [4] is an endemic medium-sized pit viper distributed mainly at altitudes above 2000 m in central Taiwan. Remarkably, the taxonomy and genus name of *T. gracilis* have been controversial [5]. Phylogenetic analyses based on the mitochondrial and nuclear gene sequences of various pitvipers revealed that *T. gracilis* is phylogenetically close to *Ovophis okinavensis* in central Ryukyu [6,7,8]. However, *T. gracilis* gives live birth to its offspring, whereas *O. okinavensis* is among the few egg-laying pit viper species. Moreover, the prey ecology of *T. gracilis* and *O. okinavensis* is rather different [9,10], and how this affects their venom proteomes remains to be further explored and clarified.

*T. gracilis* snakebites mainly elicit hemorrhagic symptoms in patients, including local tissue damage (myonecrosis, dermal necrosis, edema, hemorrhage, and blistering) and systemic coagulopathy [11], although *T. gracilis* envenoming cases are rare. To date, no specific antivenom is available for *T. gracilis* envenomation, and the venom proteome has not been resolved. *T. gracilis*-envenomed patients have been treated with local “bivalent hemotoxic-antivenom” against *Viridovipera stejnegeri* and *Protobothrops mucrosquamatus*, but this antivenom failed to relieve the local lesions of the patients before they received surgical intervention [11].

Previously, we cloned and sequenced *T. gracilis* venom proteins belonging to three major toxin families: an acidic phospholipase A_2_ (PLA_2_) and a Lys49-homolog of PLA_2_ [12], ten venom serine proteases (SVSPs) [13], and five metalloproteinases (SVMPs) including PII-class and its disintegrin (DIS) domain [14]. We have shown that the amino acid sequences of some representative toxins, including Lys49-PLA_2_, several SVSP variants, and the PI-class metalloprotease of *T. gracilis*, are highly similar to those of the corresponding venom toxins of *O. okinavensis* [15,16]. The taxonomy of *T. gracilis* and its relationship with other eastern Asian pitvipers, meanwhile, remains puzzling, and this species and those under the genus *Gloydius* have been shown to be the most likely Asian sisters of New World pitvipers [7,17,18,19]. Thus, it is not surprising that PIII-SVMP and some SVSP variants expressed in *T. gracilis* venom bear high structural similarities to the corresponding toxins from New World pitvipers [13,14].

Venomic studies may provide important insights into the pathophysiology of envenoming and the molecular evolution of venom toxin multigene families [20]. Our aim was to investigate the venom proteome and full sequences of not only the major but also the secondary or minor toxin families of *T. gracilis* in order to better understand the composition and evolution of its venom and to treat snakebites effectively. In the present study, *T. gracilis* venom proteomics were studied using high-performance liquid chromatography (HPLC) and liquid chromatography-tandem mass spectrometry (LC-MS/MS). We also cloned and sequenced the venom gland cDNAs encoding Cys-rich secretory proteins (CRISPs) and vascular endothelial growth factors (VEGFs). These *T. gracilis* toxins were compared with other pitviper toxins using a BLAST search and sequence alignments. Our results may provide a deeper understanding of the venom composition of *T. gracilis* and the evolutionary relationships between *T. gracilis* and other related pitvipers, such as the east Asian *Ovophis*, *Protobothrops*, and the New World *Crotalus*.

## 2. Results

### 2.1. Chromatographic and Electrophoretic Profiling of T. gracilis Venom

Using a C_18_ reverse-phase column, crude *T. gracilis* venom was separated into 39 peptide/protein fractions by HPLC (Figure 1A). These fractions were collected and analyzed using sodium dodecyl sulfate-polyacrylamide gel electrophoresis (SDS-PAGE) under reducing conditions (Figure 1B). The first 13 fractions (eluted within the first 55 min) failed to show any bands on the gel. Notably, most medium-to high-MW venom proteins (5000–260,000) were eluted between 55 and 130 min and collected in fractions 14–39 (Figure 1A). As expected, most of the small peptides or proteins eluted earlier than large proteins, and the basic variants of the venom toxins eluted earlier than the acidic variants of the same toxin family. However, SDS-PAGE results revealed that most of the HPLC peaks contained multiple proteins or subunits rather than a single purified protein (Figure 1B).

### 2.2. T. gracilis Venom Proteomic Analysis

Mass spectrometric analyses of the proteins in the HPLC peaks (particularly the broader ones) revealed that they may contain several proteins from different toxin families (Table 1 and Appendix A). Overall, 155 toxin proteoforms were identified in the HPLC fractions, except for peaks 1–5 and 7. Sixteen of these proteoforms were *T. gracilis* venom proteins whose full amino-acid sequences have been deduced from venom gland transcriptome, including three SVMPs, six SVSPs, two PLA_2_s, two CRISPs, one DIS, and two VEGFs (Appendix A). By measuring the peak area under the curve of the HPLC chromatogram, the relative abundances of low-molecular-weight peptides, nontoxins (e.g., keratins), and unidentified compounds in the venom together accounted for 20.1%, which were mainly from the first 13 fractions. The identified peptides/proteins were categorized into 13 toxin families and sorted quantitatively (Appendix A).

The relative abundances of individual proteins in each chromatographic fraction were calculated and consolidated, as detailed in the Materials and Methods section at the end of this paper, and the *T. gracilis* venom proteome is summarized in a pie chart (Figure 1C). Of these, SVMPs (40.3%) are the most abundant toxin family and were dominant in fractions 30–39, with a 22.5 kDa PI-SVMP eluted mainly in fractions 34–35 (Figure 2). SVSPs (15.3%) are also abundant and dominant in fractions 17–24, 28, followed by PLA_2_s (6.6%), which were mainly eluted in fractions 25–27; LAAOs (5.0%), which were dominant in fraction 29; CRISPs (4.6%), which were dominant in fractions 14 and 16; DISs (3.2%), which were derived from PII-SVMP precursors and were dominant in fraction 13; VEGFs (2.9%), which were dominant in fractions 10 and 15; and C-type lectin-like proteins (snaclecs, 1.9%), which were dominant in fraction 23 (Figure 1C and Figure 2). Less abundant toxins in *T. gracilis* venom included nerve growth factors (NGF, 0.07%), phospholipase B (PLB, 0.06%), hyaluronidases (HYA, 0.01%), 5′-nucleotidases (5′NT, 0.01%), and cystatins (<0.01%) (Appendix A).

The Tgc-SVMPs comprises PI-class (75.7%), PII-class (5.3%), and PIII-class (19.0%) enzymes, represented by 40 proteoforms, and matched to the published SVMP sequences of *T. gracilis* [14] or other species. Notably, some peptides detected in fractions 6–13 appeared to be hydrolyzed fragments of SVMPs or LAAOs, which possibly resulted from autodegradation during experimental handling of the samples. Our results also showed that 54 SVSP proteoforms were detected and partially matched previously published sequences of Tgc-SVSPs [13] or SVSPs from other species. The LAAO, CRISP, DIS, VEGF, and snaclec families have 9, 9, 2, 5, and 13 proteoforms, respectively. The PLA_2_ family is dominated by acidic PLA_2_s, with 13 proteoforms detected that partially matched with previously published sequences of Tgc-E6 [12] or acidic PLA_2_s from other species.

### 2.3. Two Cysteine-Rich Secretory Proteins (CRISPs) of T. gracilis Venom

The cDNAs encoding the two novel CRISPs (Tgc-CRa and Tgc-CRb) were cloned and fully sequenced from venom glands and submitted to GenBank under the accession numbers ACE73569.1 (gi|190195325) and ACE73570.1 (gi|190195327), respectively. Tgc-CRa and Tgc-CRb appeared to be paralogs, with only 67% sequence similarity. They were aligned with possible orthologous snake venom CRISPs retrieved using BLASTp (Figure 3A and Figure 3B, respectively). We are not able to find any venom-CRISP sequences of other *Trimeresurus* species in databanks. All the pitviper venom CRISPs contain 221 amino acid residues (Figure 3A,B), whereas those from true vipers may contain 220 residues [21]. The 16 Cys residues and sequences in their N-terminal half, including the pathogenesis-related protein-1 (PR-1) domain [22], are highly conserved. Tgc-CRa is acidic, and its sequence is 95% identical to that of Ook-CR [15] and >99% similar to the venom CRISPs of *Bothriechis schlegelii* and *Protobothrops* species (Figure 3A). In contrast, Tgc-CRb is basic and most similar to serotriflin from the blood of *P. flavoviridis* [23] and a serotriflin-like protein (i.e., Pmu-CRL) from *P. mucrosquamatus*; it is also similar to some basic CRISPs of elapid venom (Figure 3B), and these CRISPs are possibly also expressed in tissues other than venom glands. As shown in Table 1, Appendix A, Tgc-CRa and Tgc-CRb were eluted in the HPLC fractions 14 and 16, respectively (Table 1, Figure 2), and the content of Tgc-CRa was higher than that of Tgc-CRb.

### 2.4. Three VEGFs Are Expressed in the T. gracilis Venom Gland

Both the tissue and the venom-types of VEGFs have distinct biochemical properties and are common components of most viperid venom [24,25]. Here, we cloned and sequenced the cDNAs encoding three distinct VEGFs from *T. gracilis* venom glands, which were deposited to GenBank with accession numbers OQ614863–OQ614865, for Tgc-VGFa, Tgc-VGFb, and Tgc-VGFc, respectively. Their sequences were aligned with possible orthologous snake venom VEGFs retrieved from a BLAST search, respectively (Figure 4A,B). Thus far, NCBI databases do not contain any venom VEGF sequences from other *Trimeresurus* species. Apparently, the 154-residue Tgc-VGFa is identical to or >99% similar to the VEGFs expressed in the venom of *O. okinavensis*, *Protobothrops*, and some New World pitvipers, and those present in the venom of true vipers (subfamily Viperinae) (Figure 4A). Both Tgc-VGFb and Tgc-VGFc contain 122 residues and are approximately 98% similar to each other; both are 93% identical to the sequence of the Ook-VGF protein (Figure 4B) and eluted in fractions 10 and 15, respectively (Table 1, Figure 2). Both types of VEGFs contain conserved receptor-binding residues, and their C-terminal residues contain potentially basic regions responsible for binding heparin (Figure 4A,B).

## 3. Discussion

### 3.1. T. gracilis Venom Proteome

The major toxin families expressed in the venom of most pit vipers are metalloproteases, phospholipase A_2_, serine proteases, and snaclecs [26,27], which are also present in the *T. gracilis* venom. At least 12 different protein families have been identified in *T. gracilis* venom, with eight major families (SVMPs, SVSPs, PLA_2_s, LAAOs, CRISPs, DISs, VEGFs, and snaclecs) comprising 79.8% of all the venom components (Figure 1C). Additionally, low levels of NGFs, PLBs, HYAs, 5′NTs, and cystatins were identified (Appendix A). Other unidentified components of *T. gracilis* venom may mainly include low-molecular-weight peptide families, such as bradykinin-potentiating peptides, C-type natriuretic peptides, and tripeptidyl SVMP inhibitors [28]. To further verify the presence of these peptides in the *T. gracilis* venom, we searched for both trypsin-digested and non-trypsin-digested sequences by mass spectrometry using the non-redundant NCBI database. We detected two proteoforms, bradykinin-potentiating and natriuretic peptides, in fractions 4–6 by partially matching previously published sequences of *O. okinavensis* and *Bothrops atrox* (see Appendix A). Other minor venom enzymes, such as glutaminyl cyclase, aminopeptidase, and phosphodiesterase [26], are expected to be present in *T. gracilis* venom, but this has not been confirmed.

Among the 10 Tgc-SVSP variants [13], relatively high levels of Tgc-KN1, Tgc-KN4, Tgc-PAH1/2, and Tgc-PA3 were detected by proteomic analysis (Table 1; Appendix A). Among the five reported Tgc-SVMPs [14], Tgc-MP (PI class), Tgc-PIIc, and Tgc-PIII were clearly present in the venom (Table 1). In addition, two CRISP variants (Tgc-CRa and Tgc-CRb) and two venom-type VEGF variants (Tgc-VGFb and Tgc-VGFc) were also identified (Table 1). Venom VEGFs usually comprise 2–5% of the pit viper venom proteome [29], and Tgc-VGFb and VGFc contribute 2.9% of the venom proteome (Figure 1C). We previously cloned and isolated an acidic PLA_2_ (Tgc-E6W30) from Tgc venom (collected much earlier, not from Mt. Daxue) with a total yield of approximately 6% (*w*/*w*) [12], which is consistent with the relative abundance of acidic PLA_2_s in the *T. gracilis* venom proteome (6.6%; Appendix A). Other minor acidic PLA_2_ variants, or possibly an E6A30-PLA_2_, may also be present in the Tgc-venom analyzed in the present study, which could be highly similar to the acidic PLA_2_s isolated from *O. okinavensis* and *G. brevicaudus* (formerly *G. halys* or *G. blomhoffii*) (Table 1 and Appendix A). A proteoform eluted by HPLC in fraction 26 (Appendix A) was assigned as Tgc-K49 by searching on the NCBI database, but it is more likely to be an acidic PLA_2_ variant for three reasons. (1) Basic K49-PLA_2_ homolog usually eluted earlier than acidic PLA_2_s from the RP-HPLC column in 0.1% TFA, but this proteoform was eluted in fraction 26 like other acidic PLA_2_s (eluted in fractions 22–28). (2) Venom content of K49-PLA_2_ homologs is usually higher than those of the enzymatically active PLA_2_s, but this proteoform was detected only once and based on two peptides which match a mutated region (residues 70–100) in the Tgc-K49 sequence, and the region happens to be highly similar to the corresponding regions in Tgc-E6W30 and Ook-E6A30 PLA_2_s [12]. (3) The high number of acidic PLA_2_s proteoforms detected (Table 1 and Appendix A) strongly suggests the presence of more than one acidic PLA_2_ isoforms in *T. gracilis* venom and this proteoform is likely a E6A30-PLA_2_.

### 3.2. Comparison of Venom Composition and Toxicity among Closely Related Species

Tgc-MP (a PI-SVMP) is probably as hemorrhagic as okinalysin because of their 95% sequence similarity [14,16]. By acting synergistically with other venom components, abundant Tgc-MP may play a crucial role in the pathophysiology of *T. gracilis* envenoming. One of the prominent pathologies associated with the hemorrhagic PI-SVMPs is the development of extensive blistering [30], which may become a reservoir of venom toxins that can continuously damage the local tissues. Another distinct function of a number of PI-SVMPs is their ability to activate potent inflammatory responses directly [31]. Our results reveal high structural similarities between the venom proteins of *T. gracilis* and *O. okinavensis*; not only are the full sequences of their venom PLA_2_s, PI-SVMPs, CRISPs, and VEGFs most similar, but also the tryptic peptide sequences of their LAAO, PLB, and HYA match each other (Appendix A). Although *T. gracilis* is phylogenetically closely related with *O. okinavensis*, the toxicity of *O. okinavensis* venom (LD_50_ 11 μg/g mouse, via intravenous injection; [32]) is much weaker than that of *T. gracilis* venom (LD_50_ 3 μg/g mouse, via intraperitoneal injection; Tsai et al., unpublished data). The difference in their lethality could be partly explained by the differential expression of their venom toxins, i.e., SVSPs are dominant in *O. okinavensis* venom [15,33], whereas SVMPs are dominant in *T. gracilis* venom. *T. gracilis* venom promotes hemorrhage, hypotension, and impaired blood coagulation, which is consistent with mammalian predation by adult *T. gracilis*. *O. okinavensis* venom is comprised of overwhelmingly abundant SVSPs and fewer SVMPs (Figure 5), apparently representing a hybrid strategy optimized mainly for frogs [16], in addition to small mammals.

The venom proteomes of many Asian and New World pit viper species have recently been reported [27,37,38]. We are able to compare the venom proteome of *T. gracilis* with those of hemorrhagic and phylogeographically related pit viper species [7,8], as shown in Figure 5. SVMPs are the most prominent toxin family in the venom of *T. gracilis*, *G. brevicaudus*, *P. mucrosquamatus*, *C. atrox*, *C. lannomi*, and *V. stejnegeri*; however, the proportions of their PI-, PII-, and PIII-SVMPs are rather diverse (Appendix A). Snaclecs are dominant in the venom of *T. albolabris* and *T. purpureomaculatus* compared to other species in Figure 5, and the venom proteome of both arboreal species are not similar to that of *T. gracilis*. Of note, both *T. gracilis* and *C. atrox* venom are most abundant in SVMPs and SVSPs, followed by acidic PLA_2_s, while SVSPs are the most abundant family in *Ovophis* venom [15,33] that appears to lack DISs (Figure 5). It is also recognized that *T. gracilis* is the Asian sister of the New World pitvipers [6,7,8] and *T. gracilis* and *C. atrox* share high sequence similarities in their PIII-SVMPs and some SVSP variants [13,14]. Possibly because of similar diet ecology in adults, *T. gracilis* and *C. atrox* share the venom proteome with grossly similar proportions of the major toxin families (Figure 5), and their lethalities to mice are close, as LD_50_ of *C. atrox* venom is 5.0 μg/g mouse for intraperitoneal injection or 2.72 μg/g mouse for intravenous injection [39,40]. Nevertheless, the results of comparing the proteomic data from different studies may be confounded by the variations in the protein detection method, ages of the snakes, or other factors (summarized in Appendix A), and need to be explained with caution. For example, conditions of pre-treatment by trypsin could be inconsistent in the proteomic studies, in-solution tryptic digestion provided a higher number of proteins identified, and a larger sequence coverage for bottom-up proteomic studies, as compared to using in-gel digestion [41].

### 3.3. Sequence Comparison of T. gracilis Venom CRISPs

CRISPs are generally not abundant in snake venom, but are widely distributed taxonomically. The presence of CRISP toxins with high degrees of sequence similarity in all snakes suggests earlier diversification of CRISPs before the divergence between Viperidae and the remaining Colubroids [42,43]. The 19-residue signal peptides of CRISPs are highly conserved and favorable for cDNA cloning and sequencing using PCR. In the present study, two venom CRISPs, Tgc-CRa and Tgc-CRb, were fully sequenced for the first time, and a single CRISP transcript was identified in the *O. okinavensis* transcriptome (Figure 3). Tgc-CRa is highly similar to CRISPs identified in the venom of *O. okinavensis*, *Gloydius* and *Protobothrops* (Figure 3A). Their C-terminal Cys-rich domain (CRD) contained three highly conserved disulfide bridges and a proline bracket [44] (Figure 3A,B). Triflin (from *P. flavoviridis*) and ablomin (from *G. blomhoffi*) are L-type Ca^2+^-channel antagonists of arterial smooth muscle contractions that promote vasodilation and hypotension. CRISPs purified from *Bothrops* venom species may induce inflammatory responses and interfere with complement pathways, generating bioactive fragments (C3a, C4a, and C5a) and anaphylatoxins [45]. Similar to triflin, both Ook-CRa and Tgc-CRa contain hydrophobic residues at Phe^189^, Met^195^, Tyr^205^, and Phe^215^, which were shown by crystallographic studies to obstruct the target ion channels, and the highly conserved Glu^186^ and Phe^189^ are the most likely functional residues [46]. In contrast to most of the known venom CRISP sequences, an *N*-glycosylation site was present at N^48^ in serotriflin and N^44^ in Pgu-CRX2 (Figure 3B). In both Tgc-CRb and serotriflin, Phe^189^ is replaced by Tyr^189^, and the “Pro-bracket” regions 84–90 show low similarities to those of Tgc-CRa and triflin; thus, they are unlikely to bind identical ion-channels.

### 3.4. Sequence Comparison of T. gracilis Venom VEGFs

We deduced that the protein sequences of three Tgc-VEGF variants, Tgc-VGFa of 154 amino acid residues, appeared to be tissue type-specific variants and similar to human VEGF-A (Figure 4A), whereas Tgc-VGFb and Tgc-VGFc of 122 residues were snake venom types (Figure 4B), which are strongly hypotensive toxins [29]. As pointed out previously [25], the structures of tissue-type VEGFs (or VEGF-A) are highly conserved among venomous snakes and even among all vertebrates (Figure 4A), whereas those of venom-type VEGFs (also annotated as VEGF-F) are highly diversified in the regions around the receptor-binding loops and C-terminal putative coreceptor-binding regions (Figure 4B) and show different affinities to heparin [29]. Ook-VFa (AB852007.1), 154 residues long, is the most highly expressed VEGF in *O. okinavensis* venom [15]. In contrast, *T. gracilis* venom contains mainly Tgc-VGFb and Tgc-VGFc but not Tgc-VGFa (Table 1), and Tgc-VGFb is 93% identical to Ook-VFb (Figure 4B). As expected, Tgc-VGFb and Tgc-VGFc may increase vascular permeability, cause hypotension, and facilitate the spread and transport of toxin molecules, particularly when synergized with the KN subtype of SVSP [24,29].

## 4. Conclusions

Our proteomic profiling of *T. gracilis* venom was facilitated by using both comprehensive and more specific or restricted databases resulting from extensive cDNA sequencing of the three major toxin families (PLA_2_, SVSP, and SVMP) and the resolution of full sequences of *T. gracilis* venom CRISPs and VEGFs in the present study. Our results demonstrate that *T. gracilis* venom toxins qualitatively resemble those of *O. okinavensis* rather than other *Trimeresurus* and *Protobothrops* species, which is consistent with the close phylogeographic linking of *T. gracilis* and *O. okinavensis* [7,19]. The differential expression of venom proteins of *T. gracilis* and *O. okinavensis* (Figure 5) can be explained by the adaptation of both species to different environments and prey ecologies. Being an Asian sister of the New World pit vipers, *T. gracilis* retains some ancient venom genes (e.g., PIII-SVMP) that bear high sequence similarity to the corresponding toxin genes of some hemorrhagic *Crotalus* species [14]. In contrast to most *Crotalus* venom, *T. gracilis* venom lacks crotamine-like myotoxins and highly diversified PII-SVMPs, but is rich in hemorrhagic PI-SVMPs and VEGF-F. The relatively abundant Tgc-MP, Tgc-KN1 and Tgc-KN4, and Tgc-VGFb and Tgc-VGFc could explain the tissue damage, hypotension, and coagulopathy observed in *T. gracilis* envenoming. It has been demonstrated that using pan-specific effective antivenoms immunized with venoms from only a few species of pitvipers could treat the envenoming by other pitvipers if the immunizing venoms contain toxin families that are representative of the species to which the antivenom is targeted [47]. Our results suggest that antivenom prepared with stronger antigenicity against pitvipers’ PI-SVMPs could be a better candidate to treat *T. gracilis* envenoming. It is also possible to test whether the antivenom against hemorrhagic *Crotalus* venom (or adding it to the Taiwan bivalent hemotoxic-antivenom) could effectively treat *T. gracilis* envenoming. Further studies on the genome and taxonomy of *T. gracilis* and the pharmacology of its venom toxins are required to clarify its conservation status as well as its venom pathophysiology.

## 5. Materials and Methods

### 5.1. Chemicals

All the chemicals and reagents used were of analytical grade. Bovine serum albumin (BSA), formic acid (FA), dithiothreitol (DTT), and iodoacetamide (IAM) were purchased from Sigma-Aldrich (Burlington, MA, USA). Ammonium bicarbonate (AMBIC) was purchased from J.T. Baker (Phillipsburg, NJ, USA). Protein Assay dye was purchased from Bio-Rad (Hercules, CA, USA). The ExcelBand™ 3-color broad-range protein marker (5–245 kDa) was purchased from Smobio (Hsinchu, Taiwan). C_18_ RP-HPLC column (250 × 4.6 mm, 5 μm particle) was obtained from Thermo Scientific™ BioBasic™ (Waltham, MA, USA). HPLC grade acetonitrile (ACN) was purchased from Honeywell (Charlotte, NC, USA). Trifluoroacetic acid (TFA) was purchased from Acros Organics (Geel, Belgium). Sequence-grade modified trypsin was purchased from Promega (Madison, WI, USA).

### 5.2. Animals and Venom

We collected four adult *T. gracilis* (3 females and 1 male) samples from Mount Daxue, Central Taiwan, from September 2020 to December 2022. The corresponding author of this study is responsible for the taxonomic identification of snakes. According to Lin and Tu [9], snakes with a snout-vent length (SVL) larger than 22 cm were considered adults. The mean ± SEM (range) of the SVL and body mass of each collected snake was 50.3 ± 9.46 (44.0–64.0) cm and 64.0 ± 14.1 (48.0–79.0) g, respectively. Venom samples were collected manually at intervals of 14 d or more after venom collection or snake feeding. The wet venom yield per first collection of each snake was 47.0 ± 28.9 (10.7–81.2) mg. Crude venoms from the four *T. gracilis* specimens were pooled in equal proportion, lyophilized in an FD-series freeze-dryer and CES-series centrifugal evaporator (Panchum Scientific Corp., Kaohsiung, Taiwan), and stored at −80 °C until analysis.

### 5.3. Determination of Venom Protein Concentration

The lyophilized venom samples were dissolved with ultrapure water and centrifuged 10,000× *g* at 4 °C for 10 min, and the protein concentration of the supernatant was determined in triplicate using a protein assay dye (Bio-Rad, Hercules, CA, USA) with bovine serum albumin as calibration standard.

### 5.4. Reverse-Phase High-Performance Liquid Chromatography

*T. gracilis* venom containing 1.0 mg venom proteins was reconstituted in 20 μL ultrapure water and subjected to C_18_ reverse-phase fractionation using an HPLC system (Chromaster 5160 Pump and Chromaster 5410 UV detector, Hitachi, Tokyo, Japan). The C_18_ column was pre-equilibrated with 0.1% TFA in water (Buffer A) and eluted with 0.1% TFA in ACN (Buffer B) at a flow rate of 1.0 mL/min, using a linear gradient of 5% B for 5 min, 5–10% B for 5 min, 10–20% B for 30 min, 20–30% B for 5 min, 30–60% B for 90 min, and 60–70% B for 5 min. The protein elution was monitored at 215 nm and fractions were collected manually, lyophilized, and stored at −80 °C until use.

### 5.5. Sodium Dodecyl Sulfate-Polyacrylamide Gel Electrophoresis

Protein fractions collected during RP-HPLC were further analyzed by SDS-PAGE according to the method described by Laemmli [48]. The ExcelBand™ 3-color broad-range protein marker (5–245 kDa) was used as a calibration standard. Approximately 5 μg of protein from each fraction was loaded to the 12.5% polyacrylamide gel under reducing conditions and electrophorized at 110 V for 2 h. The gels were stained with Coomassie Brilliant Blue R-250 (Bio-Rad) and de-stained for visualization.

### 5.6. In-Solution Tryptic Digestion and Peptide Identification by Mass Spectrometry

After lyophilization, 10 μg of protein from each fraction was reduced with DTT, alkylated with IAM, and hydrolyzed with trypsin at an enzyme:substrate ratio of 1:25. The resulting peptides were desalted using MIcroSpin™ columns according to the manufacturer’s protocol (Cytiva, Amersham, UK). The samples were lyophilized, reconstituted in 5% ACN/0.1% FA in water, and subjected to nanoscale electrospray ionization liquid chromatography-tandem mass spectrometry (nano-ESI-LCMS/MS) using a Dionex Ultimate 3000 RSLC system (Thermo Scientific, Waltham, MA, USA) coupled with a Q Exactive mass spectrometer (Thermo Scientific). Samples were loaded in a C_18_ column (75 μm × 150 mm, 2 μm, 100 Å) (Thermo Scientific Acclaim™ PepMap™, Waltham, MA, USA) at a flow rate of 0.25 μL/min. The injection volume was 5 μL per sample and the mobile phase was 0.1% FA in water (Solution A) and 0.1% FA in 95% ACN (Solution B). The gradient applied was: 1% B for 5.5 min, 1–30% B for 39.5 min, 30–60% B for 3 min, 60–80% B for 2 min, 80% B for 10 min, 80–1% B for 5 min, and 1% B for 5 min. The ion polarity was set to positive ionization mode. Spectra were acquired in MS/MS mode with an MS scan range of 200–3000 *m*/*z* and an MS/MS scan range of 50–3000 *m*/*z*. The 10 most intense ions from the MS scan were subjected to fragmentation for MS/MS spectra. Data were analyzed with PEAKS Studio 10.5 (Bioinformatics Solutions Inc., Waterloo, ON, Canada), and the peptide-mass-finger-printing results were searched based on the non-redundant NCBI database of Serpentes (taxid:8570). Carbamidomethyl was used for static modification, and oxidation was used for dynamic modification. Protein/peptide identification was validated using the following filters: protein false discovery rate (FDR) ≥1% and unique peptides ≥1; the protein/peptide found was based on the identity of partial sequences. Keratin peptides were eliminated from further analyses. The relative abundance (%) of individual proteins in each chromatographic fraction was determined following previous methods [49,50]:Relative abundance of protein Q (%) = (mean spectral intensity of protein Q in fraction R/total mean spectral intensity in fraction R) × AUC of fraction R from HPLC (%)

The area under the curve (AUC) for each collected fraction was automatically integrated and determined from the HPLC chromatogram using Chromaster software (Hitachi, Tokyo, Japan). For each protein identified in the individual fraction, the number of spectra, the number of unique peptides, the “mean spectral intensity of protein Q in fraction R”, and “total mean spectral intensity in fraction R”, as well as other detailed data, are provided in Appendix A.

### 5.7. Molecular Cloning and Sequence Determination of CRISPs and VEGFs

*T. gracilis* venom cDNAs were prepared from venom gland mRNAs as described previously [12]. To amplify and clone the cDNAs encoding venom CRISPs, PCR was conducted using SuperTaq DNA polymerase with a pair of mixed-base oligonucleotide primers [51] designed according to the highly conserved cDNA regions in the nucleotide database. Primer 1 was designed in the sense direction: TTCA(A/C)AACA(A/G)(C/T)AGAAATG, and primer 2 was designed in the antisense direction: GATGCTACA(T/C)AG(T/G)CTTGTG [52]. DNA fragments of approximately 1.0 kb were amplified by PCR, as shown by electrophoresis of the products on a 1% agarose gel, and harvested.

The abbreviation of *T. gracilis* (Tgc) was used to name novel peptides. cDNAs encoding both the venom and serum types of Tgc-VEGFs were cloned using different sets of degenerate primers for PCR amplification. Specific primer pairs were designed based on the conserved nucleotide sequences previously used for venom VEGFs [25,53]. The PCR-amplified DNA products were analyzed on a 1% agarose gel and harvested. After treatment with polynucleotide kinase, the amplified cDNAs were inserted into the pGEM-T easy vector (Promega Corp., Madison, WI, USA) and transformed in *Escherichia coli* strain JM109. White transformants and cDNA clones were selected. The DNA Sequencing System (model 373A) and TaqDye-Deoxy terminator-cycle sequencing kit (PE Applied Biosystems, Waltham, MA, USA) were used to determine the nucleotide sequences. The protein sequences of *T. gracilis* venom CRISPs and VEGFs were deduced from their nucleotide sequences.

### 5.8. BLAST Analyses and Sequence Alignments

Protein-to-protein BLAST (BLASTp) was used to retrieve the most similar sequences for each of the novel Tgc-CRISP and Tgc-VEGF variants, using the non-redundant NCBI database (http://www.ncbi.nlm.nih.gov (accessed on 21 March 2023)). The retrieved toxin homologs were from a broad selection of pitviper genera, and preferably those have been purified and characterized. The sequences were aligned using Clustal X2 [54] and MUSCLE [55] in MEGA X [56]; gaps were introduced to optimize the comparison, and % identities or % similarities of the sequences were calculated using the sequence manipulation suite [57].

## Figures and Tables

**Figure 1 toxins-15-00408-f001:**
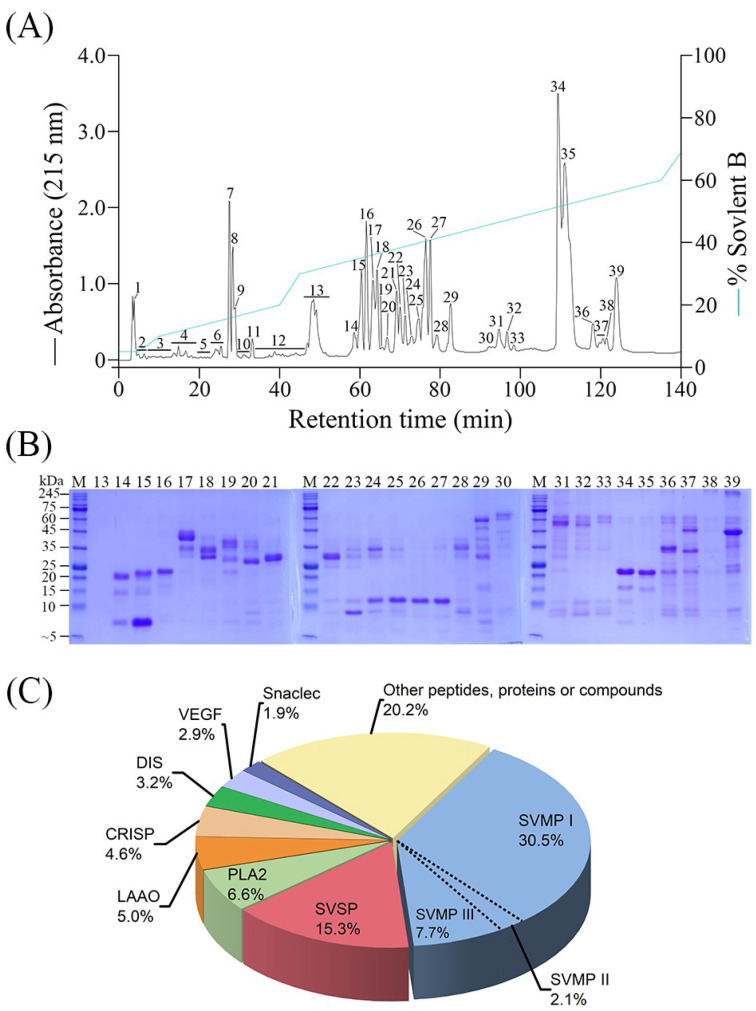
Venomic analysis of adult *Trimeresurus gracilis* from Mt. Daxue, Taiwan. (**A**) Reversed phase high performance liquid chromatography (RP-HPLC) profile of *T. gracilis* pooled venom (1.0 mg). The underline below numbers indicates that several (minor) fractions are collected together for mass spectrometry (MS) analysis. (**B**) Sodium dodecyl sulfate-polyacrylamide gel electrophoresis (SDS-PAGE) analyses of the HPLC peaks under reducing condition. Gel lanes loaded with the first 13 fractions failed to show any bands. (**C**) Pie chart representing relative abundance (in percentage of total venom components) of different toxin families based on the results of MS analysis. SVMP, snake venom metalloproteinase; SVSP, snake venom serine protease; PLA_2_, phospholipase A_2_; LAAO, L-amino acid oxidase; CRISP, cysteine-rich secretory protein; DIS, disintegrin; VEGF, vascular endothelial growth factor; snaclec, C-type lectin-like protein. The fraction “Other peptides, proteins or compounds” include nerve growth factor (0.07%), phospholipase B (0.06%), hyaluronidase (0.01%), 5′-nucleotidase (0.01%), cystatins (<0.01%), low molecular weight or small and hydrolized peptides, nontoxins (e.g., keratins), and other unidentified compounds.

**Figure 2 toxins-15-00408-f002:**
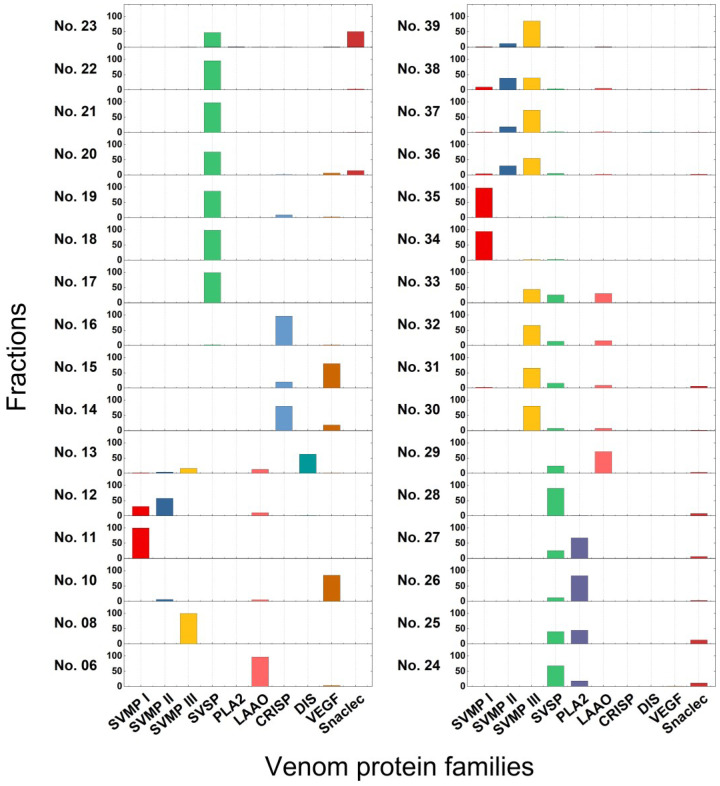
Comparisons on the relative abundance (%) of eight *Trimeresurus gracilis* toxin-families detected in the HPLC fractions. SVMP, snake venom metalloproteinase; SVSP, snake venom serine protease; PLA_2_, phospholipase A_2_; LAAO, L-amino acid oxidase; CRISP, cysteine-rich secretory protein; DIS, disintegrin; VEGF, vascular endothelial growth factor; snaclec, C-type lectin-like protein.

**Figure 3 toxins-15-00408-f003:**
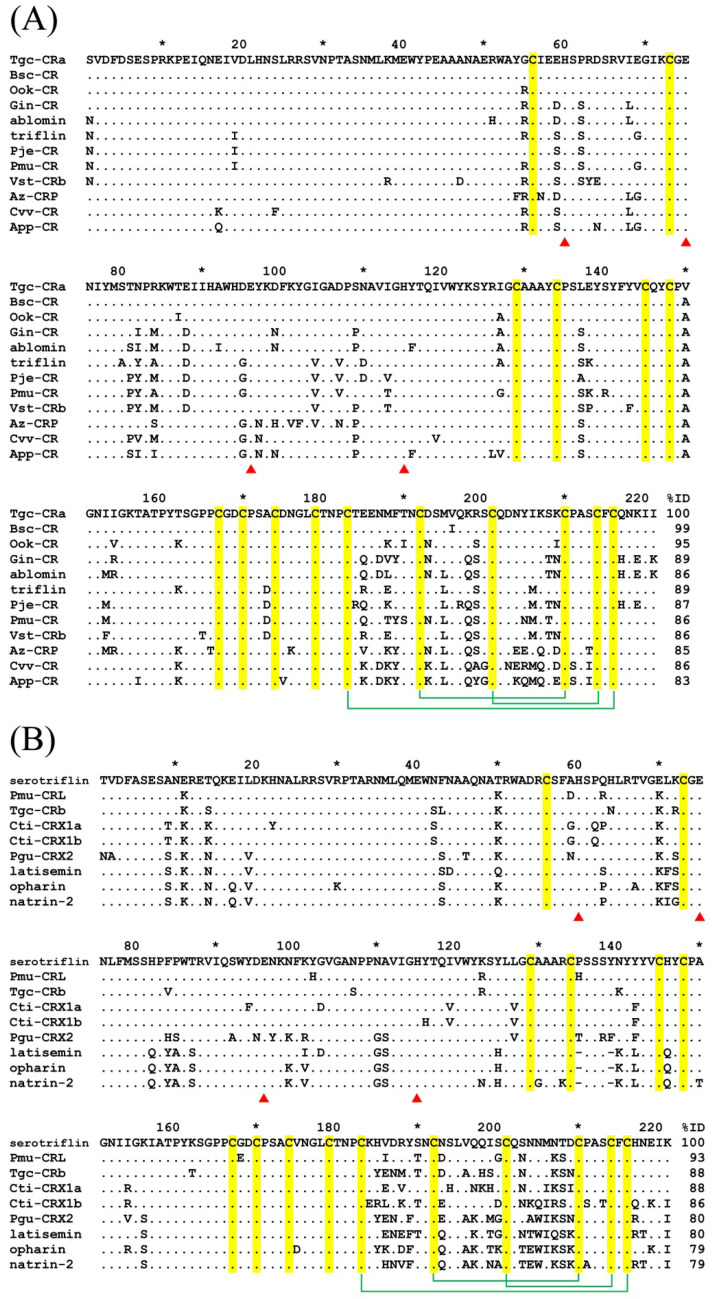
Sequence alignments of snake venom Cys-rich secretory proteins (CRISPs). Conserved Cys residues are highlighted in yellow, potential Ca^2+^ binding residues are indicated by red triangles, and gaps are shown by hyphens. * indicates a marker count from the average of adjacent numbers. Three pairs of C-terminal disulfide bridges are shown in green. (**A**) The acidic CRISP homologs retrieved by BLAST. Accession numbers and species are: Tgc-CRa, ACE73569.1; Bsc-CR, ACE73559.1 (*Bothriechis schlegelii*); Ook-CR, BAN82147.1 (*Ovophis okinavensis*); Gin-CR, UQT19685.1 (*Gloydius intermedius*); ablomin, UQT19680.1 (*G. blomhoffii*); triflin, Q8JI39.1 (*Protobothrops flavoviridis*); Pje-CR, Q7ZZN9.1 (*P. jerdonii*); Pmu-CR, XP_015678374.1 (*P. mucrosquamatus*); Vst-CRb, ACE73573.1 (*Viridovipera stejnegeri*); Az-CRP, ACE73558.1 (*Azemiops feae*); Cvv-CR, ACE73566.1 (*Crotalus v. viridis*); and App-CR, Q7ZTA0.1 (*Agkistrodon p. piscivorus*). (**B**) The basic CRISPs and CRISPs homologous to Tgc-CRb. Accession numbers and species are: Serotriflin, P0CB15 (*P. flavoviridis*); Pmu-CRL, XP_015678372 (*P. mucrosquamatus*); Tgc-CRb, ACE73570.1; Cti-CRX1a, XP_039185543.1 (*C. tigris*); Cti-CRX1b, XP_039185562.1 (*C. Tigris*); Pgu-CRX2, XP_034288380.1 (*Pantherophis guttatus* blood); latisemin, Q8JI38.1 (*Laticauda semifasciata*); opharin, ACN93671.1 (*Ophiophagus hannah*); and natrin-2, Q7ZZN8.1 (*Naja atra*).

**Figure 4 toxins-15-00408-f004:**
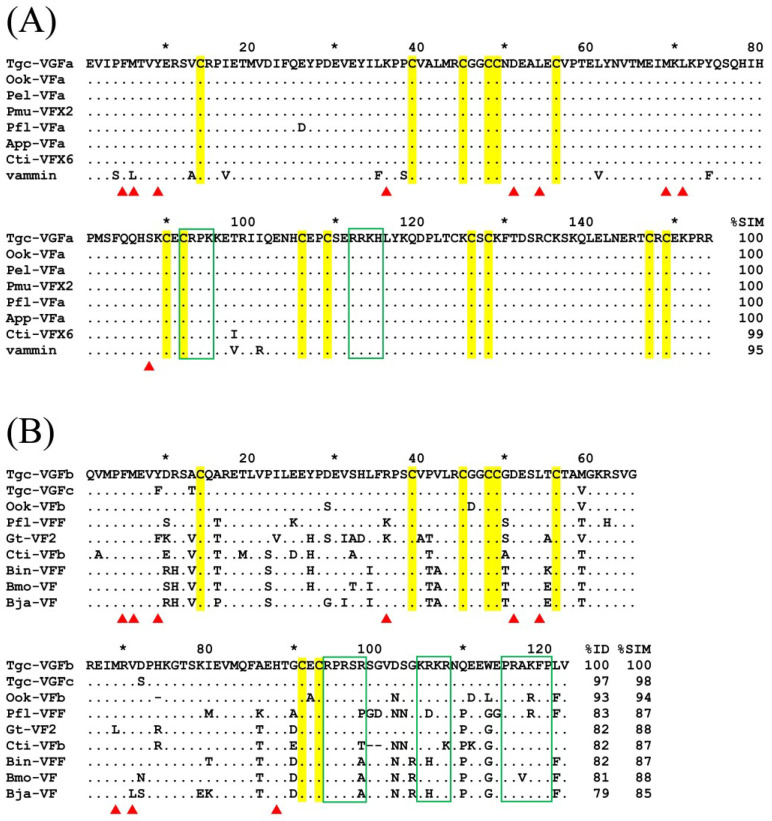
Sequence alignments of VEGF family proteins expressed in viperid venom glands. Conserved Cys residues are highlighted in yellow, amino acid residues potentially involved in VEGF-receptor-binding are marked with red triangles below the sequences, and possible heparin-binding regions are boxed with green lines. * indicates a marker count from the average of adjacent numbers. (**A**) Tgc-VGFa and homologs retrieved by BLASTp. Accession numbers and species are: Tgc-VGFa, OQ614863; Ook-VFa, BAN89442.1 (*Ovophis okinavensis*); Pel-VFa, BAP39940.1 (*Protobothrops elegans*); Pmu-VFX2, XP_015673445.1 (*P. mucrosquamatus*); Pfl-VFa, BAD38845.1 (*P. flavoviridis*); App-VFa, C0K3N4.1 (*Agkistrodon p. piscivorus*); Cti-VFX6, XP_039198673.1 (*Crotalus tigris*); and Vammin, C0K3N5.1 (*Vipera ammodytes ammodytes*). (**B**) Tgc-VGFb and Tgc-VGFc homologs retrieved by BLASTp. Accession numbers and species are: Tgc-VGFb, OQ614864; Tgc-VGFc, OQ614865; Ook-VFb, BAN82145.1 (*O. okinavensis*); Pfl-VFF, P67862.1 (*P. flavoviridis*); Gt-VF2, BAO57712.1 (*Gloydius tsushimaensis*); Cti-VFb, XP_039218052.1 (*Crotalus tigris*); Bin-VFF, Q90X24.1 (*Bothrops insularis*); Bmo-VF, ATU85531.1 (*B. moojeni*); and Bja-VF, KAG5858117.1 (*B. jararaca*).

**Figure 5 toxins-15-00408-f005:**
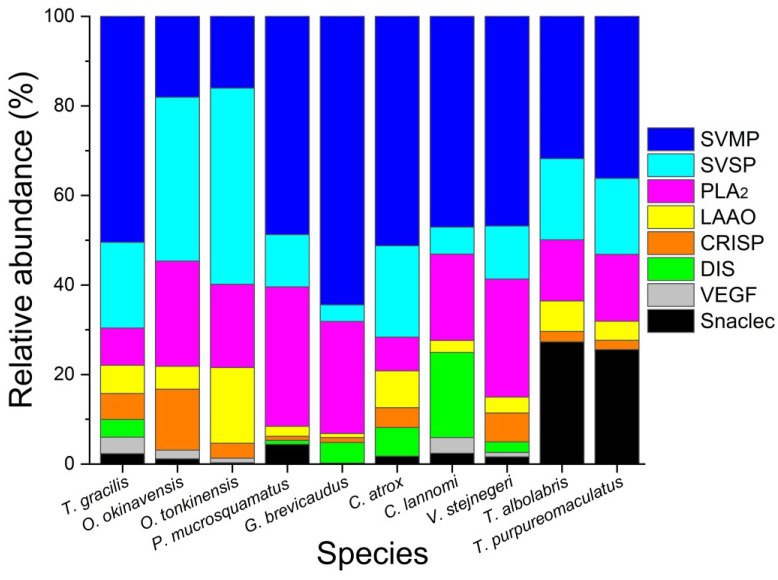
Comparison of the venom proteome of *Trimeresurus gracilis* to those of other related pitvipers. The abundance of eight key toxin families (relative to the sum of their total abundances) in each venom species were calculated based on published data, respectively: *Ovophis okinavensis* and *O. tonkinensis* [33], *Gloydius brevicaudus* [34], *Protobothrops mucrosquamatus* and *Viridovipera stejnegeri* [35], *Crotalus atrox* [36] and *C. lannomi* [37], *T. albolabris*, and *T. purpureomaculatus* [38]. SVMP, snake venom metalloproteinase; SVSP, snake venom serine protease; PLA_2_, phospholipase A_2_; LAAO, L-amino acid oxidase; CRISP, cysteine-rich secretory protein; DIS, disintegrin; VEGF, vascular endothelial growth factor; snaclec, C-type lectin-like protein.

**Table 1 toxins-15-00408-t001:** *Trimeresurus gracilis* (Tgc) venom proteome as profiled by reversed phase high performance liquid chromatography (RP-HPLC) and nanoscale electrospray ionization liquid chromatography-tandem mass spectrometry (nano-ESI-LC-MS/MS). Minor components (<0.1% of total venom components) are not displayed.

HPLC Fraction /Toxin Family	Protein (Proteoform) Name	Database Accession (NCBI)	Species	Protein Score	Relative Abundance (%)
**Fraction 6**					
LAAO	L-amino oxidase	gi|538260091	*Ovophis okinavensis*	69.70	1.91
**Fraction 8**					
SVMP III	Metalloprotease PIII [Tgc-PIII] *	gi|335892636	*Trimeresurus gracilis*	34.41	0.11
**Fraction 10**					
VEGF	Tgc-VGFb	OQ614864	*Trimeresurus gracilis*	75.04	0.28
**Fraction 11**					
SVMP I	Metalloprotease PI [Tgc-MP]	gi|335892630	*Trimeresurus gracilis*	50.20	0.63
**Fraction 12**					
SVMP II	Metalloprotease precursor H4, partial	gi|7340946	*Deinagkistrodon acutus*	82.92	0.91
SVMP I	Metalloprotease PI [Tgc-MP]	gi|335892630	*Trimeresurus gracilis*	71.95	0.49
LAAO	Chain A Amine oxidase	gi|1186227927	*Bothrops atrox*	132.93	0.16
**Fraction 13**					
DIS	Metalloprotease PIIb [gracilisin] **	gi|335892632	*Trimeresurus gracilis*	99.53	3.13
LAAO	Chain A Amine oxidase	gi|1186227927	*Bothrops atrox*	93.75	0.65
SVMP III	P-III_metalloprotease	gi|547223066	*Ovophis okinavensis*	202.13	0.31
SVMP III	Metalloproteinase type III 12b	gi|1041577317	*Agkistrodon conanti*	183.99	0.22
SVMP II	Snake venom metalloprotease precursor	gi|2035122236	*Bothrops jararaca*	127.91	0.17
SVMP III	Metalloprotease PIII [Tgc-PIII]	gi|335892636	*Trimeresurus gracilis*	147.80	0.17
SVMP III	Metalloproteinase (type III) 1a	gi|818935191	*Crotalus adamanteus*	130.40	0.11
**Fraction 14**					
CRISP	Serotriflin	gi|1002598708	*Protobothrops mucrosquamatus*	190.38	0.46
CRISP	Cysteine-rich seceretory protein Og-CRPb, partial [Tgc-CRb]	gi|190195327	*Trimeresurus gracilis*	347.51	0.28
VEGF	Tgc-VGFb	OQ614864	*Trimeresurus gracilis*	207.39	0.21
CRISP	CRiSP-Sut-27	gi|476539526	*Suta fasciata*	52.48	0.12
**Fraction 15**					
VEGF	Tgc-VGFc	OQ614865	*Trimeresurus gracilis*	121	1.16
VEGF	Tgc-VGFb	OQ614864	*Trimeresurus gracilis*	195.33	0.80
CRISP	Cysteine-rich seceretory protein Dr-CRPK	gi|190195321	*Daboia russelii*	155.56	0.23
VEGF	Cadam10_VEGF-1	gi|1178170176	*Crotalus adamanteus*	85.2	0.20
CRISP	Cysteine-rich seceretory protein Og-CRPa [Tgc-CRa]	gi|190195325	*Trimeresurus gracilis*	287.81	0.18
CRISP	Cysteine-rich secretory protein, partial	gi|2205501413	*Malpolon monspessulanus*	109.71	0.13
**Fraction 16**					
CRISP	Cysteine-rich seceretory protein Dr-CRPK	gi|190195321	*Daboia russelii*	176.09	0.88
CRISP	Cysteine-rich seceretory protein Bs-CRP	gi|190195305	*Bothriechis schlegelii*	364.81	0.69
CRISP	Cysteine-rich secretory protein, partial	gi|2205501413	*Malpolon monspessulanus*	128.48	0.61
CRISP	Cysteine-rich secretory protein TRI1	gi|123898155	*Trimorphodon biscutatus*	53.33	0.26
**Fraction 17**					
SVSP	Serine proteinase 12a	gi|1180525223	*Agkistrodon contortrix contortrix*	118.97	1.34
SVSP	Thrombin-like enzyme LmrSP-3	gi|1714612439	*Lachesis muta rhombeata*	59.05	0.75
SVSP	Ancrod=thrombin-like alpha-fibrinogenase	gi|247212	*Akistrodon rhodostoma*	89.78	0.56
SVSP	Venom thrombin-like enzyme, partial	gi|118430266	*Deinagkistrodon acutus*	109.05	0.34
**Fraction 18**					
SVSP	Thrombin-like enzyme collinein-4	gi|1109550140	*Crotalus durissus collilineatus*	97.45	0.39
SVSP	Serine proteinase 8b	gi|1041578893	*Sistrurus tergeminus*	120.13	0.39
SVSP	Thrombin-like enzyme bhalternin; Fibrinogen-clotting enzyme	gi|298351882	*Bothrops alternatus*	107.85	0.32
SVSP	Snake venom serine protease pallase	gi|158514815	*Gloydius halys*	147.7	0.22
SVSP	Ancrod-like protein	gi|1334616	*Calloselasma rhodostoma*	104.21	0.20
SVSP	Serine endopeptidase	gi|1333445426	*Crotalus tigris*	157.2	0.18
SVSP	Serine proteinase 2	gi|1041577225	*Agkistrodon conanti*	120.95	0.18
SVSP	Agkihpin	gi|484358552	*Gloydius halys*	125.98	0.16
SVSP	Plasminogen-activator subtype serine protease (PA1/2) [Tgc-PAH1/2]	gi|2289393718/2289393720	*Trimeresurus gracilis*	304.56	0.14
SVSP	Snake venom serine protease precursor	gi|2035122138	*Bothrops jararaca*	172.04	0.14
**Fraction 21**					
SVSP	Kallikrein-like serine protease (KN4) [Tgc-KN4]	gi|2289393712	*Trimeresurus gracilis*	277.88	0.74
SVSP	Kallikrein-like serine protease (KN1) [Tgc-KN1]	gi|2289393706	*Trimeresurus gracilis*	156.88	0.65
SVSP	Serine proteinase 1	gi|1041577231	*Agkistrodon conanti*	130.59	0.17
**Fraction 22**					
SVSP	Kallikrein-like serine protease (KN4) [Tgc-KN4]	gi|2289393712	*Trimeresurus gracilis*	219.86	0.50
SVSP	Kallikrein-like serine protease (KN1) [Tgc-KN1]	gi|2289393706	*Trimeresurus gracilis*	151.81	0.43
SVSP	Serine proteinase 1	gi|1041577231	*Agkistrodon conanti*	104.59	0.23
SVSP	Snake venom serine protease serpentokallikrein-2 isoform X1	gi|1002585685	*Protobothrops mucrosquamatus*	147.93	0.14
**Fraction 23**					
Snaclec	C-type lectin LmsL; Galactose-specific lectin; Mutina	gi|34922643	*Lachesis stenophrys*	220.07	0.22
Snaclec	Galactose binding lectin	gi|538260107	*Ovophis okinavensis*	204.87	0.16
Snaclec	Galactose binding lectin, partial	gi|538259813	*Protobothrops flavoviridis*	168.47	0.16
Snaclec	Chain B Galactose-specific lectin	gi|33357350	*Crotalus atrox*	208.84	0.14
SVSP	Serine proteinase 8c	gi|1041578891	*Sistrurus tergeminus*	103.02	0.12
**Fraction 24**					
SVSP	Kallikrein-like serine protease (KN1) [Tgc-KN1]	gi|2289393706	*Trimeresurus gracilis*	172.27	0.12
SVSP	Plasminogen-activator subtype serine protease (PA3) [Tgc-PA3]	gi|2289393722	*Trimeresurus gracilis*	154.61	0.11
**Fraction 25**					
SVSP	Kallikrein-like serine protease (KN1) [Tgc-KN1]	gi|2289393706	*Trimeresurus gracilis*	130.75	0.27
PLA_2_	Acidic phospholipase A_2_ [Tgc-E6]	gi|59727071	*Trimeresurus gracilis*	266.65	0.23
SVSP	Thrombin-like enzyme	gi|38146946	*Gloydius shedaoensis*	108.67	0.14
SVSP	Thrombin-like enzyme halystase	gi|3122187	*Gloydius blomhoffii*	100.35	0.14
PLA_2_	Phospholipase A_2_ precursor	gi|743759444	*Protobothrops tokarensis*	66.77	0.13
PLA_2_	Acidic phospholipase A_2_	gi|129420	*Gloydius blomhoffii*	117.08	0.13
PLA_2_	Phospholipase A_2_ type IIE	gi|384110782	*Dispholidus typus*	83.97	0.13
SVSP	Serine proteinase 19b	gi|1041577233	*Agkistrodon conanti*	112.35	0.12
PLA_2_	Phospholipase A_2_	gi|584481356	*Ovophis makazayazaya*	92.74	0.11
**Fraction 26**					
PLA_2_	Acidic phospholipase A_2_ [Tgc-E6]	gi|59727071	*Trimeresurus gracilis*	444.11	1.70
PLA_2_	Phospholipase A_2_ isozyme CTs-A3, partial	gi|37785867	*Viridovipera stejnegeri*	136.6	1.08
PLA_2_	Phospholipase A_2_, partial	gi|538259861	*Protobothrops flavoviridis*	147.33	0.90
SVSP	Kallikrein-like serine protease (KN4) [Tgc-KN4]	gi|2289393712	*Trimeresurus gracilis*	135.89	0.25
SVSP	Kallikrein-like serine protease (KN1) [Tgc-KN1]	gi|2289393706	*Trimeresurus gracilis*	143.16	0.19
**Fraction 27**					
PLA_2_	Acidic phospholipase A_2_ [Tgc-E6]	gi|59727071	*Trimeresurus gracilis*	300.99	0.79
PLA_2_	Phospholipase A_2_, partial	gi|538259861	*Protobothrops flavoviridis*	121.88	0.53
PLA_2_	Phospholipase A_2_ isozyme CTs-A3, partial	gi|37785867	*Viridovipera stejnegeri*	117.83	0.40
SVSP	Kallikrein-like serine protease (KN1) [Tgc-KN1]	gi|2289393706	*Trimeresurus gracilis*	102.54	0.20
SVSP	Kallikrein-like serine protease (KN4) [Tgc-KN4]	gi|2289393712	*Trimeresurus gracilis*	121.04	0.17
SVSP	Plasminogen-activator subtype serine protease (PA3) [Tgc-PA3]	gi|2289393722	*Trimeresurus gracilis*	144	0.15
**Fraction 28**					
SVSP	Plasminogen-activator subtype serine protease (PA3) [Tgc-PA3]	gi|2289393722	*Trimeresurus gracilis*	137.28	0.28
SVSP	Serine proteinase 19b	gi|1041577233	*Agkistrodon conanti*	108.99	0.27
SVSP	Plasminogen-activator subtype serine protease (PA1/2) [Tgc-PAH1/2]	gi|2289393718/2289393720	*Trimeresurus gracilis*	134.51	0.19
SVSP	Kallikrein-like serine protease (KN4) [Tgc-KN4]	gi|2289393712	*Trimeresurus gracilis*	102.15	0.17
SVSP	Kallikrein-like serine protease (KN1) [Tgc-KN1]	gi|2289393706	*Trimeresurus gracilis*	96.4	0.13
**Fraction 29**					
LAAO	L-amino-acid oxidase	gi|347602330	*Vipera ammodytes ammodytes*	212.56	0.66
LAAO	Chain A Ahp-laao	gi|48425312	*Gloydius halys*	199.24	0.51
LAAO	L-amino acid oxidase	gi|538260091	*Ovophis okinavensis*	275.32	0.42
SVSP	Kallikrein-like serine protease (KN4) [Tgc-KN4]	gi|2289393712	*Trimeresurus gracilis*	111.55	0.16
SVSP	Plasminogen-activator subtype serine protease (PA3) [Tgc-PA3]	gi|2289393722	*Trimeresurus gracilis*	138.9	0.13
LAAO	BATXLAAO1	gi|1127252627	*Bothrops atrox*	176.6	0.12
SVSP	Kallikrein-like serine protease (KN1) [Tgc-KN1]	gi|2289393706	*Trimeresurus gracilis*	119.14	0.12
**Fraction 30**					
SVMP III	Metalloproteinase, partial	gi|297593822	*Echis carinatus sochureki*	50.52	0.31
SVMP III	Metalloproteinase-disintegrin-like atrolysin-A, partial	gi|1663479917	*Protobothrops mucrosquamatus*	183.98	0.25
SVMP III	BATXSVMPIII16	gi|1127252547	*Bothrops atrox*	74.01	0.11
**Fraction 31**					
SVMP III	Metalloprotease PIII [Tgc-PIII]	gi|335892636	*Trimeresurus gracilis*	307.13	0.46
SVMP III	Metalloproteinase-disintegrin-like atrolysin-A, partial	gi|1663479917	*Protobothrops mucrosquamatus*	236.99	0.40
**Fraction 32**					
SVMP III	Metalloprotease PIII [Tgc-PIII]	gi|335892636	*Trimeresurus gracilis*	202.57	0.37
SVMP III	Metalloproteinase-disintegrin-like atrolysin-A, partial	gi|1663479917	*Protobothrops mucrosquamatus*	162.6	0.12
LAAO	L-amino acid oxidase	gi|538260091	*Ovophis okinavensis*	181.72	0.11
**Fraction 34**					
SVMP I	P-II_metalloprotease ***	gi|547223068	*Ovophis okinavensis*	217.81	5.79
SVMP I	Metalloprotease PI [Tgc-MP]	gi|335892630	*Trimeresurus gracilis*	362.99	3.88
SVMP I	P-II metalloprotease, partial ***	gi|547223015	*Protobothrops flavoviridis*	189.92	0.20
SVMP III	MDC-6d	gi|1829138061	*Crotalus atrox*	167.81	0.22
SVSP	Kallikrein-like serine protease (KN4) [Tgc-KN4]	gi|2289393712	*Trimeresurus gracilis*	165.77	0.11
**Fraction 35**					
SVMP I	Metalloprotease PI [Tgc-MP]	gi|335892630	*Trimeresurus gracilis*	360.09	19.26
SVSP	Kallikrein-like serine protease (KN1) [Tgc-KN1]	gi|2289393706	*Trimeresurus gracilis*	119.16	0.15
SVSP	Kallikrein-like serine protease (KN4) [Tgc-KN4]	gi|2289393712	*Trimeresurus gracilis*	105.69	0.11
**Fraction 36**					
SVMP III	Metalloproteinase, partial	gi|1773624963	*Dispholidus typus*	62.96	0.38
SVMP II	Tgc-PIIc	OK482650	*Trimeresurus gracilis*	308.73	0.10
SVMP II	Metalloproteinase type II 6c	gi|1041579264	*Sistrurus miliarius barbouri*	127.08	0.10
**Fraction 37**					
SVMP II	Snake venom metalloprotease precursor	gi|2035122167	*Bothrops jararaca*	106.31	0.16
SVMP II	Tgc-PIIc	OK482650	*Trimeresurus gracilis*	206.13	0.12
SVMP III	Metalloproteinase type III 9b	gi|1041579226	*Sistrurus miliarius barbouri*	75.23	0.11
**Fraction 38**					
SVMP II	Tgc-PIIc	OK482650	*Trimeresurus gracilis*	220.3	0.11
**Fraction 39**					
SVMP III	Metalloprotease P-III 3, partial	gi|675402421	*Protobothrops flavoviridis*	137.47	0.76
SVMP III	Metalloproteinase (type III) 6a	gi|1180525232	*Agkistrodon contortrix contortrix*	222.51	0.68
SVMP III	Metalloproteinase type III 12b	gi|1041577317	*Agkistrodon conanti*	206.17	0.60
SVMP III	P-III_metalloprotease	gi|547223066	*Ovophis okinavensis*	318.96	0.51
SVMP II	Snake venom metalloprotease precursor	gi|2035122236	*Bothrops jararaca*	111.17	0.42
SVMP III	Metalloproteinase type III 5a	gi|1041579244	*Sistrurus miliarius barbouri*	230.38	0.33
SVMP III	Snake venom metalloprotease precursor	gi|2035122126	*Bothrops jararaca*	115.29	0.19

* Updated toxin names are indicated in square brackets. ** The sequence matches gracilisin [14]. *** The sequence matches Tgc-MP [14] and okinalysin [16].

## Data Availability

Data are contained within the article or Appendix A.

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
