# Peer review of "Venom Proteomics of Trimeresurus gracilis, a Taiwan-Endemic Pitviper, and Comparison of Its Venom Proteome and VEGF and CRISP Sequences with Those of the Most Related Species"

_toxins, 2023, doi:10.3390/toxins15070408_

Round 1
Reviewer 1 Report
The authors conducted a proteomic study of Trimeresurus gracilis venom to reveal its composition, and compared it with the venom of other pitvipers closely related in phylogeny. The work is meaningful in developing suitable antivenom and understanding adaptive evolution of Asian pitvipers. However, some more information should be provided to help readers to understand their conclusions.
1. Overall, at least 15 distinct Tgc venom proteins and 124 proteoforms were identified. The 124 proteoforms seem to be the 124 toxins listed in Table S2, but what are these 15 venom proteins? It is better to add this information in the manuscript as well.
2. How did authors choose which peak area of the HPLC to measure to calculate the relative abundance of the low-molecular-weight peptides, body proteins, and unidentified compounds? Some more explanation of the procedure is needed. In the text, the proportion of these compounds in the venom is 17.4% , but it is 17.6 % in the Figure 1C. Please confirm this.
3. Authors compared the sequences of CRISP and VEGF in Tgc venom with that of other pitvipers except for species in Trimeresurus genus. How similar are the CRISP and VEGF in Tgc venom to that of other Trimeresurus species? The information should be provided, and this is necessary to support the conclusion in line 323-325.
4. In the text, authors mentioned the detection of a venom protein named Tgc-VGFa. Which sequence is Tgc-VGFa in Table 1 and supplementary tables? Besides, does Tgc_VF correspond to Tgc-VGF? Please make these consistent.
5. In section 3.2, authors compared venom proteomes of different ptivipers. As some proteomic data are from other studies, in which the protein detection method and snake ages may vary, are the data still comparable? Besides, the data from published data is said to be recalculated. What is the procedure? Above information is worth mentioning in the method part.
Author Response
The authors conducted a proteomic study of Trimeresurus gracilis venom to reveal its composition, and compared it with the venom of other pitvipers closely related in phylogeny. The work is meaningful in developing suitable antivenom and understanding adaptive evolution of Asian pitvipers. However, some more information should be provided to help readers to understand their conclusions.
- Overall, at least 15 distinct Tgc venom proteins and 124 proteoforms were identified. The 124 proteoforms seem to be the 124 toxins listed in Table S2, but what are these 15 venom proteins? It is better to add this information in the manuscript.
Ans: We have changed the words “at least 15 distinct Tgc venom proteins…”. The numbers of proteoforms have been recalculated (lines 17, 117, 155, 159, and 161-162), and there were 155 toxin proteoforms that belong to 13 viperid venom toxin families. Sixteen of these proteoforms matched the venom proteins whose full amino-acid sequences have been deduced from T. gracilis venom gland cDNAs. In addition, we also corrected a few mistakes of calculation in lines 19, 21-22, 111, 124, 143-144, 149, 151, 154, and 234, and the corresponding data in all figures and tables.
- How did authors choose which peak area of the HPLC to measure to calculate the relative abundance of the low-molecular-weight peptides, body proteins, and unidentified compounds? Some more explanation of the procedure is needed. In the text, the proportion of these compounds in the venom is 17.4% , but it is 17.6 % in the Figure 1C. Please confirm this.
Ans: Our MS/MS results suggested that the low-molecular-weight peptides, nontoxins, and unidentified compounds were mainly in the first 13 fractions, which could be deduced from Supplementary Table S1. The original value of 17.6 % was calculated by adding up the relative abundance of nontoxins and unidentified components (17.4%) and those of identified minor toxins, including nerve growth factor (0.07%), phospholipase B (0.06%), hyaluronidase (0.01%), 5’-nucleotidase (0.01%), and cystatins (< 0.01%). Because we corrected a few mistakes of calculation, the value “17.6%” has now been corrected to “20.2%” (line 22 and Figure 1C), which are the total relative abundances of minor toxins, nontoxins, and unidentified peptides or compounds. The value “17.4%” has now been corrected to “20.1%” (line 124).
- Authors compared the sequences of CRISP and VEGF in Tgc venom with that of other pitvipers except for species in Trimeresurus genus. How similar are the CRISP and VEGF in Tgc venom to that of other Trimeresurusspecies? The information should be provided, and this is necessary to support the conclusion in line 323-325.
Ans: We could not find any venom CRISP and VEGF sequences in databanks for other Trimeresurus species, as explained in lines 170-171 and 205-206. It is noticeable that T. gracilis is phylogenetically linked with O. okinavensis rather than other Trimeresurus species [ref. 6-8].
- In the text, authors mentioned the detection of a venom protein named Tgc-VGFa. Which sequence is Tgc-VGFa in Table 1 and supplementary tables? Besides, does Tgc-VF correspond to Tgc-VGF? Please make these consistent.
Ans: Tgc-VGFa was not detected by LC-mass spectrometry, thus did not appear in Table 1 or supplementary tables. Tgc-VF is the same to Tgc-VGF and for consistency we have changed all “Tgc-VF” to “Tgc-VGF” in the text.
- In section 3.2, authors compared venom proteomes of different ptivipers. As some proteomic data are from other studies, in which the protein detection method and snake ages may vary, are the data still comparable? Besides, the data from published data is said to be recalculated. What is the procedure? Above information is worth mentioning in the method part.
Ans: Regarding this matter, we have discussed more (lines 319-325, i.e. the results of comparing the proteomic data from different studies may be confounded by the variations in the protein detection method, snake ages, or other factors, and need to be explained with caution). The procedure to recalculate the data has been added to Figure 5 legend.
Reviewer 2 Report
The article describes the proteome of a venom pool of four adult snakes Trimeresurus gracilis using two complementary approaches RP-HPLC and peptide identification by mass spectrometry of a tryptic digestion of the venom solution. The venom of this viper showed a high amount of metalloproteases and serine proteases, followed by acidic PLA2, LAAO, CRISP, disintegrins, and snaclecs. In addition, it describes two minor toxins in more detail – VEGF and CRISP. Treatment of accidents with T. gracilis snake has been done with antivenom against Viridovipera stejnegeri and Protobothrops mucrosquamatus, which seems ineffective against the local effects of envenomation.
The manuscript deserves to be published for describing the characterization of the crude venom of T. gracilis that is little known; in addition analyzes toxins not previously studied in detail, such as CRISPs and VEGF.
Minor considerations
It is interesting to discuss more how the results of proteome could improve the treatment of T. gracilis bite, as stated in the “key contribution” (L27-28) and introduction (L65-66).
Could this proportionality of toxins between T. gracilis and C. atrox be due to diet or habitat or envenoming symptomatology? Clustering the characteristics of the venoms of related snake species and including the proportion of different classes of metalloproteases could illustrate the relationship among them and indicate the antivenom to be added to the bivalent antivenom already used to treat the local symptoms. This approach was used in Sousa LF, Nicolau CA, Peixoto PS, Bernardoni JL, Oliveira SS, et al. (2013) Comparison of Phylogeny, Venom Composition and Neutralization by Antivenom in Diverse Species of Bothrops Complex. PLoS Negl Trop Dis 7(9): e2442. doi:10.1371/journal.pntd.0002442.
Material and methods: were venoms collected from 4 adult female or male snakes?
It is not possible to access the reference below:
32. LD50men. Available online: https://web.archive.org/web/20120413182323/
http://www.venomdoc.com/LD50/LD50men.html 542 (accessed on 20 May 2023).
Author Response
The article describes the proteome of a venom pool of four adult snakes T. gracilis using two complementary approaches RP-HPLC and peptide identification by mass spectrometry of a tryptic digestion of the venom solution. The venom of this viper showed a high amount of metalloproteases and serine proteases, followed by acidic PLA2, LAAO, CRISP, disintegrins, and snaclecs. In addition, it describes two minor toxins in more detail – VEGF and CRISP. Treatment of accidents with T. gracilis snake has been done with antivenom against Viridovipera stejnegeri and Protobothrops mucrosquamatus, which seems ineffective against the local effects of envenomation. The manuscript deserves to be published for describing the characterization of the crude venom of T. gracilis that is little known; in addition analyzes toxins not previously studied in detail, such as CRISPs and VEGF.
It is interesting to discuss more how the results of proteome could improve the treatment of T. gracilis bite, as stated in the “key contribution” (L27-28) and introduction (L65-66).
Ans: Yes, we have inserted a sentence to “key contribution”: Our results suggest that antivenom prepared with stronger antigenicity against pitvipers’ PI-SVMPs should be a better choice to treat T. gracilis envenoming. Related discussion has also been added at lines 380-387.
Could this proportionality of toxins between T. gracilis and C. atrox be due to diet or habitat or envenoming symptomatology? Clustering the characteristics of the venoms of related snake species and including the proportion of different classes of metalloproteases could illustrate the relationship among them and indicate the antivenom to be added to the bivalent antivenom already used to treat the local symptoms. This approach was used in Sousa LF, Nicolau CA, Peixoto PS, Bernardoni JL, Oliveira SS, et al. (2013) Comparison of Phylogeny, Venom Composition and Neutralization by Antivenom in Diverse Species of Bothrops Complex. PLoS Negl Trop Dis 7(9): e2442. doi:10.1371/journal.pntd.0002442.
Ans: Thank you for providing the reference concerning using pan-specific effective antivenoms to treat Bothrops envenomations in South America. T. gracilis is the Asian sisters of the New World Crotalus and share high sequence similarities of their PIII-SVMPs and some SVSP variants; Fig. 5 shows that T. gracilis and C. atrox adults share the venom proteome with grossly similar proportions of the major toxin families, probably because of their similar diet ecology. We have discussed in lines 380-387 that antivenom prepared with stronger antigenicity against pitvipers’ PI-SVMPs could be a better choice to treat T. gracilis envenoming. It should be studied whether the antivenom against Crotalus venom (or adding it to the Taiwan bivalent antivenom) could effectively treat T. gracilis envenoming.
Material and methods: were venoms collected from 4 adult female or male snakes?
Ans: The venoms were collected from 3 adult females and 1 adult male, which have been added in line 403.
It is not possible to access the reference below: 32. LD50men. Available online: https://web.archive.org/web/20120413182323/ http://www.venomdoc.com/LD50/LD50men.html 542 (accessed on 20 May 2023).
Ans: The correct address shown in line 636 should be: https://web.archive.org/web/20120413182323/http://www.venomdoc.com/LD50/LD50men.html
Reviewer 3 Report
This manuscript contains an in depth look at the venom of the endemic Taiwan viper. Variability of snake venom is important aspect for all for treatment and anti-venom development. This paper provides a standard exploration of venom using proteomics and relating it to toxin families and is clearly written. Protocols are appropriate used for the study. I recommend for publication after addressing a few minor issues as outlined below.
Introduction
It is not convention to shorten a species name to an abbreviation (Tgc), T. gracilis should be always used as the abbreviation, or the full name if at the beginning of the sentence. This should be done to avoid confusion with other species and genus which can easily be shortened to Tgc, or if the taxonomic placement is changed in the future there is no miscommunication as to the species that was the focus of the study, it will also help people search manuscripts of that species. This also applies to shortening the other species to Ook.
I do appreciate that this abbreviation (Tgc) is later used to name novel peptides found in the venom, which needs to be explicitly stated in the naming convention of novel peptides in the methods.
Additionally, it is good practice cite the author of the species e.g. Trimeresaurus gracilis Oshima, 1920 and Oshima is cited in the references. This needs to occur on the first use, citing a species author is like citing a protocol.
I am a little unclear as to the background of the phylogenetic confusion – why is it confusing and which snake is it confused with? Was it the Ook snake?
Methods
Was one of the authors responsible for the taxonomic identification? Whoever was responsible for identification should be stipulated in the methods along with the species definition.
You need to explicitly state that the peptides found and sorted are based on sequence similarity – I am sure not every peptide was tested for function to be certain that the function they possess is correctly aligned to the family or if indeed they are bioactive. Homology infers that structure and function are in alignment and been completed, sequence similarity is a batter term to use and make it clear to the readers the ones that have been tested and the ones that are classified on similarity alone.
The correct abbreviation for ammonium bicarbonate is AMBIC
Results
Irrespective that TABLE 1 includes the results I fell it is distracting from your results and may be better in the Supp Material and a smaller table containing the two families you have elected to work on in the main text e.g CRISPs and VEGF’s. Additionally, you present a figure of results in the discussion (Figure 5) which should be in the results section.
I think if you colour code your sequence alignments it would make it easier for the reader to view and to darken the boxes of where you think the receptors bind. In addition, I could not find in the methodology how you to predicted binding residues?
More out of interest but the different lengths of the peptide families 280 vs 281 have these lengths been proven in any structural biology studies i.e. that the last residue ‘K’ not cleaved off during folding?
Discussion
See above comment regarding Figure 5
Be careful of such statements as L302 that they probably don’t block ion-channels – until a peptide is characterised its all hypothesis. It might be better to say “unlikely” then a definite do not.
Author Response
This manuscript contains an in depth look at the venom of the endemic Taiwan viper. Variability of snake venom is important aspect for all for treatment and anti-venom development. This paper provides a standard exploration of venom using proteomics and relating it to toxin families and is clearly written. Protocols are appropriate used for the study. I recommend for publication after addressing a few minor issues as outlined below.
It is not convention to shorten a species name to an abbreviation (Tgc), T. gracilis should be always used as the abbreviation, or the full name if at the beginning of the sentence. This should be done to avoid confusion with other species and genus which can easily be shortened to Tgc, or if the taxonomic placement is changed in the future there is no miscommunication as to the species that was the focus of the study, it will also help people search manuscripts of that species. This also applies to shortening the other species to Ook. I do appreciate that this abbreviation (Tgc) is later used to name novel peptides found in the venom, which needs to be explicitly stated in the naming convention of novel peptides in the methods.
Ans: Agree; we have changed Tgc to T. gracilis, and Ook to O. okinavensis throughout the manuscript. The abbreviation of T. gracilis (Tgc) was used to name novel peptides, as stated in line 479.
Additionally, it is good practice cite the author of the species e.g. Trimeresaurus gracilis Oshima, 1920 and Oshima is cited in the references. This needs to occur on the first use, citing a species author is like citing a protocol.
Ans: Agree, and we have cited the author of the species (Trimeresurus gracilis) in line 48.
I am a little unclear as to the background of the phylogenetic confusion – why is it confusing and which snake is it confused with? Was it the Ook snake?
Ans: It is widely agreed that Trimeresurus gracilis is not phylogenetically linked with other Trimeresurus which is arboreal or semi-aboreal. The generic status of T. gracilis and O. okinavensis is unclear. They probably belong to another genus [ref. 5]. We have added this reference in line 50.
Was one of the authors responsible for the taxonomic identification? Whoever was responsible for identification should be stipulated in the methods along with the species definition.
Ans: The corresponding author of this paper is responsible for the snake taxonomic identification (see lines 404-405).
You need to explicitly state that the peptides found and sorted are based on sequence similarity – I am sure not every peptide was tested for function to be certain that the function they possess is correctly aligned to the family or if indeed they are bioactive. Homology infers that structure and function are in alignment and been completed, sequence similarity is a batter term to use and make it clear to the readers the ones that have been tested and the ones that are classified on similarity alone.
Ans: We have stated in lines 456-457 that the protein/peptide found was based on the identity of partial sequences.
The correct abbreviation for ammonium bicarbonate is AMBIC.
Ans: It is corrected in line 394.
Irrespective that TABLE 1 includes the results I fell it is distracting from your results and may be better in the Supp Material and a smaller table containing the two families you have elected to work on in the main text e.g CRISPs and VEGF’s. Additionally, you present a figure of results in the discussion (Figure 5) which should be in the results section.
Ans: The detailed proteomic data have been shown in supplementary Table S1 and S2, and we have significantly shortened it by presenting only the proteoforms with relative abundances > 0.1% of the total venom. Figure 5 is kept in in the Discussion because most of the data are from literatures and used for comparing how different these pitvipers’ venom proteomes are.
I think if you colour code your sequence alignments it would make it easier for the reader to view and to darken the boxes of where you think the receptors bind. In addition, I could not find in the methodology how you to predicted binding residues?
Ans: We have colorized the sequence alignments in Figures 3 and 4 in the revised manuscript. The residues potentially involved in VEGF-receptor-binding were based on references in [ref. 24].
More out of interest but the different lengths of the peptide families 220 vs 221 have these lengths been proven in any structural biology studies i.e. that the last residue ‘K’ not cleaved off during folding?
Ans: We do not know the answer by now. The sequences aligned in Fig. 3B were retrieved from databanks, which usually are deduced from the cDNA sequences; any post-translational modification or processing of the venom proteins remains to be investigated.
See above comment regarding Figure 5.
Ans: Most data compared in Figure 5 are not data acquired from our study and we are keeping this figure in the Discussion.
Be careful of such statements as L302 that they probably don’t block ion-channels – until a peptide is characterised its all hypothesis. It might be better to say “unlikely” then a definite do not.
Ans: We have revised the statements in line 348.
Round 2
Reviewer 1 Report
The authors have resolved all my questions.